

# Schema and content aware classification for predicting the sources containing an answer over *corpus* and knowledge graphs

Somayeh Asadifar[1], Mohsen Kahani[1] and Saeedeh Shekarpour[2]

[1] Faculty of Engineering, Ferdowsi University of Mashhad, Mashhad, Khorasan Razavi, Iran
[2] College of Arts and Sciences: Computer Science, University of Dayton, Dayton, Ohio, United States

## ABSTRACT

Today, several attempts to manage question answering (QA) have been made in three separate areas: (1) knowledge-based (KB), (2) text-based and (3) hybrid, which takes advantage of both prior areas in extracting the response. On the other hand, in question answering on a large number of sources, source prediction to ensure scalability is very important. In this paper, a method for source prediction is presented in hybrid QA, involving several KB sources and a text source. In a few hybrid methods for source selection, including only one KB source in addition to the textual source, prioritization or heuristics have been used that have not been evaluated so far. Most methods available in source selection services are based on general metadata or triple instances. These methods are not suitable due to the unstructured source in hybrid QA. In this research, we need data details to predict the source. In addition, unlike KB federated methods that are based on triple instances, we use the behind idea of mediated schema to ensure data integration and scalability. Results from evaluations that consider word, triple, and question level information, show that the proposed approach performs well against a few benchmarks. In addition, the comparison of the proposed method with the existing approaches in hybrid and KB source prediction and also QA tasks has shown a significant reduction in response time and increased accuracy.

## INTRODUCTION

An ideal question answering (QA) system is a system that can display a response to the user within a reasonable time. The exploiting answer poses further challenges if it requires a hybrid approach (*Papadaki, Tzitzikas & Spyratos, 2020*), meaning not all questions are answered by structured or unstructured sources alone. Considering example question $Q_1$ in Table 1, the first sub-question, $sq_1$, is mapped to the query (?x, dbo:musicComposer, ?uri) which searches encyclopedic information for the music composer of the film. Thus, the answer to this query is extractable from the LOD cloud (DBPedia). On the other hand, the second sub-question, $sq_2$, specifically its phrase, "the early life of", is not encyclopedic information and so cannot be extracted from the LOD cloud. As a result, the

Corresponding author
Mohsen Kahani, kahani@um.ac.ir

**Table 1 Examples of questions. $sq_i$ and $sq_i$-triple stand for i-th sub-question and related its triple pattern, respectively.**

$Q_1$ = "Who composed the music for the film that depicts the early life of Jane Austen?"
$sq_1$ = "What film depicts the early life of Jane Austen?"
$sq_1$-triple= (? film, depicts, the early life of Jane Austen)
film = "Becoming Jane"
$sq_2$ = "Who composed the music for the film?"
$sq_2$-triple = (Who, composed the music, film)
answer = "Adrian_Johnston".

$Q_2$ = "Which writers had influenced the philosopher that refused a Nobel Prize?"
$sq_1$ = "Which philosopher refused a Nobel Prize?"
$sq_1$-triple = (? philosopher, refused, Nobel Prize)
philosopher = "Jean-Paul Sartre"
$sq_2$ = "Which writers had influenced the philosopher?"
$sq_2$-triple = (? writers, had influenced, philosopher)
answer = "Gustave_Flaubert".

sub-question, $sq_2$, should be searched using unstructured sources. Thus, to meet requirements in real situations, QA interfaces must aggregate information from both KB dataset (knowledge graph (KG)) and the corpus. Recently, much research has been conducted in this area, but all of these studies have utilized only one predetermined dataset to extract the answer (*Dimitrakis, Sgontzos & Tzitzikas, 2020*; *Park et al., 2015*; *Sun et al., 2018*). Therefore, in this category of tasks, the prediction and selection of the source is not necessary task before starting to query.

The current work conducts a hybrid search of a large number of structured datasets and unstructured corpus. The proposed method, hybridSP (hybrid Source Prediction), is the first solution to predict the data source in hybrid QA, involving several KG sources and a text source. HybridSP examines the question from two views of question words and question triple. The first view considers the question as a sequence of words. Question level information is used to determine the answer source in two categories: structured or unstructured, while triple level information is used to determine the structured source containing the answer. Only a handful of hybrid QA systems refer to the choice of source from the two sources of knowledge graph and textual source. These systems have used prioritization (*Park et al., 2015*) between these two sources or simple heuristics (*Usbeck et al., 2015*) to select the relevant source. But so far, no evaluation has been provided on execution time and accuracy for source selection. In addition, in these systems, resources are limited and pre-determined.

Among the sources of the knowledge graph (KG), source selection methods have been proposed, regardless of its application, in the topic of data integration that is represented in three main approaches: materialization (*Papadaki, Tzitzikas & Spyratos, 2020*; *Papadaki, Spyratos & Tzitzikas, 2021*), mediation (*Farré, Varga & Almar, 2019*; *Ekaputra et al., 2017*), and virtual (*Endris et al., 2017*; *Algosaibi, 2021*). The mediation approach, among these approaches, creates a balance between the cost of storage and time response. This method establishes an interface based on a mediated schema (*Farré, Varga & Almar, 2019*; *Ekaputra et al., 2017*; *Yousfi, El Yazidi & Zellou, 2020*) for dataset mapping.

On the other hand, discovering the relevant dataset on the topic of data integration falls into three categories: metadata-based (*Neto et al., 2016*; *Yumusak et al., 2017*), content-based (*Mountantonakis, 2018*), and link-based (*Ristoski, Bizer & Paulheim, 2015*). Content-based data integration systems are also divided into instance-based (*Diefenbach et al., 2019*), schema-based (*Khan, Bhowmick & Bonchi, 2017*; *Scherp & Blume, 2021*) and hybrid categories (*Mountantonakis, 2018*). In the proposed method, due to the presence of unstructured source, general metadata is not sufficient to identify the source. According to the example question $Q_2$ in the Table 1, the information about "Jean-Paul Sartre" is present in the structured semantic dataset "DBpedia", while the property with the concept of "refused" is not available and must be extracted from the unstructured data source. Therefore, the answer to the sub-question $sq_1$ should be extracted from the unstructured data source.

Since most datasets have a very large number of triples, searching at the instance level takes a lot of time. Therefore, based on the fact that in any structured dataset, the number of triples at schema level is always much less than the number of triples at the instance level, so the method proposed in this paper for predicting the source is based on the schema of the datasets. Among the existing methods, the most similar to the hybridSP used to select the source in the QA process is LODsyndesis (*Mountantonakis, 2018*). The method of this system is to focus on entity-based triples and consider the data source structure as separate indices of class, property, and entity, independently in each data source. These two factors have led to slowing down as well as low accuracy for complex questions, especially multi-hop questions in LODsyndesis.

According to the above, in the proposed method, with the aim of increasing the speed of predicting the relevant source, the data integration method based on mediated schema is used. Taking the idea from (*Ekaputra et al., 2017*) about the usage of multiple schemas for data integration, the current work builds the two levels data integration framework for dataset prediction. Schema matching against schemas is perform using hMatcher (*Yousfi, El Yazidi & Zellou, 2020*).

On the other hand, with the increase in the size and number of LOD datasets, they became a field of interest. In general, these systems can be divided into two general categories: (1) semantic parsing methods (*Zheng et al., 2018*; *Shin & Lee, 2020*; *Dimitrakis et al., 2020*), that lack sufficient flexibility, and (2) neural network methods (*Wang & Ling, 2019*; *Su & Yu, 2020*), encrypting questions and sources in vector space, which have led to increased accuracy and speed.

Similar to existing hybrid systems, KBQA systems often perform on a limited number of predetermined datasets. QAnswer (*Diefenbach et al., 2019*; *Diefenbach et al., 2020*) and LODQA (*Dimitrakis et al., 2020*) on the other hand, are KBQA systems implemented on a substantial portion of LOD. Lack of proper source selection method has been introduced as a challenge in QAnswer (*Diefenbach et al., 2019*; *Diefenbach et al., 2020*). LODQA (*Dimitrakis et al., 2020*) uses the LODsyndesis system to select the source containing the question entity. LODQA is a semantic parsing method based on Bag-of-Words. In this research, it has been used as a basic system. To increase the efficiency, the BERT-based QA (*Su & Yu, 2020*) system has been used along with the proposed system to select the

appropriate source. BERT (*Devlin et al., 2019*) is a powerful method that has emerged as the latest version of neural network methods.

The effect of adding the proposed method for source selection along with QAnswer during response extraction is also investigated.

Due to the KG datasets integration based on the middle schema in the proposed method, it is necessary to extract the question schema to search for the answer source. Recently, the issue of sequence tagging (*Panchendrarajan, 2018*) has been considered for various domains. The present research for question schema generation has been influenced by the sequence tagging work (*Mrhar et al., 2020*). That study merges two well-known sequence tagging approaches, LSTM and CRF, in a type extraction module. The current study extracts the type of subject and object of the given question (question schema) by referencing the author's idea in *Mrhar et al. (2020),* but does so using the BI-LSTM-CRF approach presented in *Yavuz et al. (2016)* over BERT vectors to increase the performance (*Elgendy, 2019*).

The present paper's main contribution is to provide a solution to predict the source (s) containing the response from a number of structured data sources and an unstructured data source. Other contributions are as follows:

- Using question words information and question structure to determine the source containing the answer from the KB sources and a corpus.
- Using the idea of data integration based on creating a mediated schema to predict the data source (s).
- Extraction of the question schema by the usage of sequence tagging according to the CRF and LSTM approach.
- Investigation of linguistic properties of questions with answers in the text by clustering BERT embedding vectors.
- Comparison of the proposed method based on different settings on several benchmarks.
- Comparison in the proposed data integration method with the state-of-the-art method for selecting the appropriate KB source on simple and multi-hop questions.
- Comparison of accuracy and speed of existing KBQA systems implemented on a large number of KB sources using the proposed data integration method.

The rest of the current paper is organized as follows. "Related Works" reviews the related works, while "Preliminaries" introduces the necessary preliminaries. The proposed approach is presented in "Approach". "Experiments" provides the empirical results and closes with concluding remarks and future works.

## RELATED WORKS

The literature related to the approach of the current research is spread over two areas: (i) hybrid question answering systems and (ii) data integration, question answering and sequence tagging in LOD. The following provides a brief overview of the state-of-the-art of these areas.

## Hybrid question answering systems

Hybrid QA approaches consist of multiple structured and unstructured sources to search for an answer. Recently, some hybrid QA works have developed a collaborative strategy. In text source annotated with KB entities, *Park et al. (2015)* employed SPARQL queries if the text strategy is not successful. Other works leverage text to enrich the vocabulary of the structured knowledge (*Xiong, Wang & Wang, 2021*; *Kartsaklis, Pilehvar & Collier, 2018*) or leverage KB facts to enrich the unstructured source (*Xiong, Wang & Wang, 2021*; *Flisar & Podgorelec, 2018*). Recent systems operate on text and KB continuously (*Sun et al., 2018*; *Sun, Bedrax-Weiss & Cohen, 2019*), such as turn the data from various types of sources into a special global representation and perform an expanded approach using memory networks (*Dimitrakis, Sgontzos & Tzitzikas, 2020*; *Miller et al., 2016*). The major weaknesses of this work which lack the rich relationship framework between information and text excerpts are solved by GRAFT-Nets (*Sun et al., 2018*) and its next system PullNet (*Sun, Bedrax-Weiss & Cohen, 2019*).

The critical point in all hybrid QA systems is that all are based on utilizing one structured source and one unstructured source for extracting the desired response. To extract answers, existing hybrid query systems do not provide a process for selecting the appropriate resource from multiple existing (structured or unstructured) sources.

## Data integration, question answering and sequence tagging in LOD

**Data Integration:** Regarding query processing in LOD, another critical issue is data integration. LOD provides a set of datasets to make finding a response and data connectivity easier. Therefore, the need for data integration is absolutely essential, because datasets are constructed independently and according to different principles.

There are three main approaches for data integration (*Mountantonakis & Tzitzikas, 2019*). The first approach is materialization (*Schuetz, Schausberger & Schrefl, 2018*), in which datasets are physically collected in the same place, thus increasing the response speed. However, many costs are incurred to incorporate the data as well as to update possible variations. The second approach is mediation (*Farré, Varga & Almar, 2019*; *Ekaputra et al., 2017*), in which one or more schema or ontology is regarded as the interface between the user and the distributed datasets. A query is first executed on the interface and then on the selected datasets. In mediation, the response rate is lower than that of the materialization method, but there is no cost for collecting data. The third approach is the virtual method (*Mountantonakis & Tzitzikas, 2020*), which is federated query processing featuring two categories of SPARQL endpoints (*e.g.*, LODatio (*Mountantonakis & Tzitzikas, 2020*)) and LDF clients (linked data fragment) (*Endris et al., 2017*).

Unlike materialization, the virtual approach does not need to pay for the datasets' warehouse storage costs nor does it need to use schemas and ontologies for the interface. However, the virtual response rate is far less than that of materialization and mediation (*Endris et al., 2017*). Therefore, the mediation method generates a balance between cost and execution time. Thus the present research selects the mediation method.

Data integration approaches, on the other hand, fall into three general categories: metadata-based (*Neto et al., 2016*; *Yumusak et al., 2017*), content-based (*Mountantonakis, 2018*), and link-based (*Ristoski, Bizer & Paulheim, 2015*), with the goal of selecting the appropriate source (*Mountantonakis & Tzitzikas, 2019*). Another hot field of research at the present time that is close to the field of data integration is called graph summarization. The purpose of this field is to calculate a meaningful and concise summary of key information in a graph (*Khan, Bhowmick & Bonchi, 2017*; *Scherp & Blume, 2021*). This goal is very similar to the motivation of this research, with the difference that our goal is to gather data on summaries obtained from several graphs.

In existing data integration approaches, metadata includes descriptions of data sources, statistics, and general information. Because the text source is present next to the KB datasets, metadata-based methods such as *Neto et al. (2016)*, *Yumusak et al. (2017)* are not applicable in the current research. The reason for this is that the answer with the same subject may not be present in the KG but can be found in the text. Content-based systems are also divided into instance-based (*Diefenbach et al., 2019*), schema-based (*Khan, Bhowmick & Bonchi, 2017*; *Scherp & Blume, 2021*) and hybrid categories (*Mountantonakis, 2018*).

The triples of each structured dataset in LOD are added at the instance level based on a specific schema. Since most datasets have a very large number of triples, searching at the instance level takes a lot of time. Therefore, based on the fact that in any structured dataset, the number of triples at schema level is always much less than the number of triples at the instance level, so the method proposed in this paper for predicting the source is based on the schema of the datasets.

Among the existing methods, the most similar to the hybridSP used to select the source in the QA process is LODsyndesis (*Mountantonakis, 2018*). This system is entity-based and extracts the sources containing the entity provided within the LOD. This method is slow because of the need to search for instance level triples to determine the source containing the response. In addition, for each source, it focuses on its triples, entities, properties and classes separately, so it performs well only on simple questions and has poor performance on multi-hop questions that are highly regarded in research today. In the present study, LODsyndesis has been used as a baseline for comparison with the proposed system.

On the other hand, previous research has shown that ontology (*Ekaputra et al., 2017*) or schema-based (*Farré, Varga & Almar, 2019*; *Yousfi, El Yazidi & Zellou, 2020*) methods handle the challenges of the mediation data integration approaches with flexibility and simplicity (*Farré, Varga & Almar, 2019*). Because ontology includes the classification of classes, their relations and instances, the current work chooses schemas for the simplicity of data integration in dataset selection.

Since authors in *Ekaputra et al. (2017)* use multiple ontologies for data integration, the present study takes that idea to manage multiple datasets. The current work addresses multiple schemas with hMmatcher's schema matching (*Yousfi, El Yazidi & Zellou, 2020*) approach, which has a higher accuracy than other methods.

**Knowledge-based question answering**: With the increase in the size and number of datasets in the LOD, research on improving the efficiency of systems in information processing as well as increasing the speed and accuracy of response has expanded. With this explanation, the existing systems fall into two general categories. In the first category, the systems are based on the semantic analysis of the given question, which do not have enough flexibility (*Zheng et al., 2018*; *Shin & Lee, 2020*; *Dimitrakis et al., 2020*). The second category increases the efficiency of the system in finding accurate answers by encrypting questions and KG in vector space (*Wang & Ling, 2019*; *Su & Yu, 2020*). In the second category, the powerful BERT method is introduced as a recent representation, which is used with a transformer structure and in a fine-tuned manner. The strength of BERT has been evaluated in research over traditional methods (*Gonz & Eduardo, 2005*). Therefore, using the rich semantic properties of BERT word embedding will increase system performance.

Although much research has been conducted in the context of QA in LOD (*Das et al., 2018*; *Qiu et al., 2020*), there is still a very small number of systems which execute queries over a large number of datasets. QAnswer (*Diefenbach et al., 2019*; *Diefenbach et al., 2020*) and LODQA (*Dimitrakis et al., 2020*) are systems, which recently implemented on a significant part of LOD. In QAnswer (*Diefenbach et al., 2019*; *Diefenbach et al., 2020*), scalability is not addressed and the challenge is to choose the correct datasets. The LODQA (*Dimitrakis et al., 2020*) system uses the LODsyndesis (*Mountantonakis, 2018*) data integration system to select the appropriate source. LODsyndesis is a content-based method (hybrid of schema and instance levels) to select the appropriate source. As stated in the data integration related works section, LODsyndesis (*Mountantonakis, 2018*) has been used in this research as a basic system in the evaluation section. In this research, the prediction of the source containing the answer is considered. Therefore, the use of the proposed system in the response extraction process is also considered. The LODQA (*Dimitrakis et al., 2020*) system, which uses LODsyndesis (*Mountantonakis, 2018*), is also considered as one of the basic systems for performing evaluations. LODQA (*Dimitrakis et al., 2020*) is a semantic analysis method, so to increase the efficiency of this research, a powerful BERT-based system (*Su & Yu, 2020*) is used along with the proposed source method in the evaluation section.

**Sequence tagging:** Lately, sequence tagging problems have attracted much attention. Recent work has featured long-term dependency learning employing Long Short-Term Memory networks (LSTM). Effective use can be made of past and future characteristics for individual positions (*via* backward and forward states) in the bidirectional LSTM network (*Panchendrarajan, 2018*). Another way to utilize tag information and generate greater tagging accuracy is applying the Conditional Random Field (CRF) for the usage of sentence level data. Inspired by *Mrhar et al. (2020)*, the current work uses CRF with the LSTM method to identify the type of subject and the object of the given question. The present study for question schema generation, adopts the method introduced by *Yavuz et al. (2016)*, with the BI-LSTM-CRF approach.

The current paper intends to examine the gap between these two considered related work areas. This is achieved by introducing a continuous and simultaneous prediction

method for selecting sources that contain the answer among multiple KB datasets and textual data sources.

## PRELIMINARIES

This section presents the necessary preliminaries. To promote the importance of the KB scheme in the current approach, the general definition of the database schema is first formally presented, followed by the definition of the KB dataset and its schema. Throughout the present paper, the term dataset refers to KB databases and the term source signifies both the KB and corpus databases. In addition, the knowledge graph (KG) and the knowledge-based (KB) dataset are used interchangeably.

### Database schema

The term database schema refers to the structure defined in a formal language which organizes the database and maintains its integrity. In other words, data is arranged within a schema and, overall, the schema is the main body of the database that determines how the database is to be constructed.

### RDF

A Resource Description Framework (RDF) is a metadata model specified in the W3C RDF recommendation (https://www.w3.org/TR/rdf-concepts/#section-Graph-syntax) for building an infrastructure of machine-readable semantics for data on the web. In the RDF model, the universe to be modeled is a set of resources, that is, essentially anything that has a universal resource identifier, URI. The RDF graph, $G$, is a set of triples with a structure (*i.e.*, subject, predicate, object) abbreviated as $(s, p, o)$[1] $\in (U \cup B) \times (U) \times (U \cup B \cup L)$, in which $U$ represents URI references, $L$ stands for literals, and $B \subset (U \cup L)$ is for blank nodes[2]. The predicate, $p$, is a relation between $s$ and $o$. Each triple is represented by the labeled graph, $s \xrightarrow{p} o$. The RDF database is named Knowledge Graph (KG).

### RDF schema (KG schema)

RDF Schema $S$ is abbreviated as RDFS, RDF(S), RDF-S or RDF/S, associated with the RDF graph, $G$, which is the built-in vocabulary, $voc(G) \subset U \cup L$, has an inheritance of classes, $C$, and certain properties, $P$, and provides basic elements for the ontology definition. Properties, denoted as a predicate, describe a relation between subject $s$ and object $o$ resources. $S$ is a set of triples $(sc, p, oc)$, in which $sc$ indicates the class of the subject and $oc$ represents the class of the object for the triple, whose predicate is property $p$. The proposed method involves two types of connections: Property-based connections and ontology-based connections. The first involves connections that have a property between two different classes (*e.g.*, dbo:PopulatedPlace, dbo:officialLanguage, dbo:Language). The latter involves the connections that exist between the various elements in an ontology. For example, the relationships "subclass-of" and "part-of" can be mentioned (See Fig. 1).

Figure 2 illustrates three triples: (i) (dbr:Italy, dbo:officialLanguage, dbr:Italian_language), (ii) (dbr:Italy, rdf:type, dbo:PopulatedPlace), and (iii) (dbr:Italian_language, rdf:type, dbo:Language). In the second triple, "dbo:PopulatedPlace" expresses the class of the resource, dbr:Italy, and, in the third triple, dbo:Language is the class of the resource,

[1] This definition is inspired by RDF triple.

[2] This feature is useful when representing complex attributes, such as address.

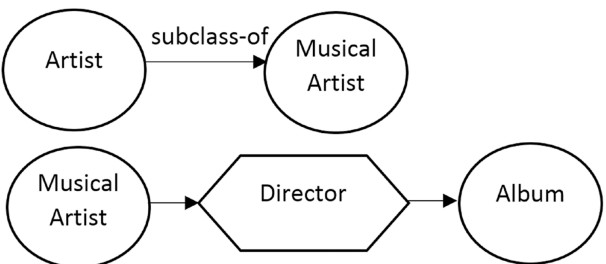

**Figure 1** Top: property-based relations and bottom: standard connections such as subsumption (*e.g.*, subclass-of) and mereology (*e.g.*, part-of).

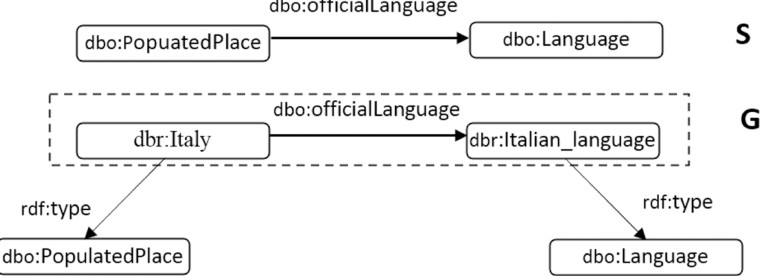

**Figure 2** (G) A piece of RDF graph and (S) related schema for the triple in the dashed line rectangle.

dbr:Italian_language. The related schema for the first triple in the RDF graph, *G*, is represented in Schema *S*, while the property, dbo:officialLanguage, describes a relation between the subject resource associated with the class of dbo:PopulatedPlace and the object resource associated with the class of dbo:Language.

The following is a formal definition of the question schema concept in the current paper.

## Question schema

The schema of the triple $(s, p, o)$ is defined as $(sc, p, oc)$, in which $sc$ represents the class of subject $s$ and $oc$ stands for the class of object $o$ by considering an underlying ontology. Since the topology and vocabulary of background knowledge graphs are heterogeneously declared in various domains, the direction of the extracted schema depends on the background schema or ontology. Therefore, in order to have a generic model, both directions of an extracted schema, $(sc, p, oc)$ and $(oc, p, sc)$, are considered.

## Baseline systems

Since in this study the proposed data integration method, hybridSP, is compared with LODsyndesis (*Mountantonakis, 2018*), the indices generated in this method are shown in Fig. 3. The main purpose of LODsyndesis is to identify data sources containing the desired entity. Thus, entity, property, constant, class, and entity-based triple indices are generated.

Entity-based Triple Index

| Entity | Property | Value | Dataset IDs |
|---|---|---|---|
| E1_Greco | LivedAt | Spain | 4 |
| | Carried out by | View Of Toledo | 1,2 |
| Heraklion | took place at | El-Greco-Birth | 1 |
| | population | 15,1324 | 1,4 |

Entity Index

| Entity | Dataset IDs |
|---|---|
| El_Greco | 1,2,4 |
| Heraklion | 1 |

Property Index

| Property | Dataset IDs |
|---|---|
| took place at | 1,2 |
| LivedAt | 3,4 |

Literal Index

| Literal | Dataset IDs |
|---|---|
| 151,324 | 1,4 |
| 1542 | 2 |

Class Index

| Class | Dataset IDs |
|---|---|
| Person | 1,2 |
| production | 3 |

**Figure 3 LODsyndesis indices (*Mountantonakis, 2018*).**

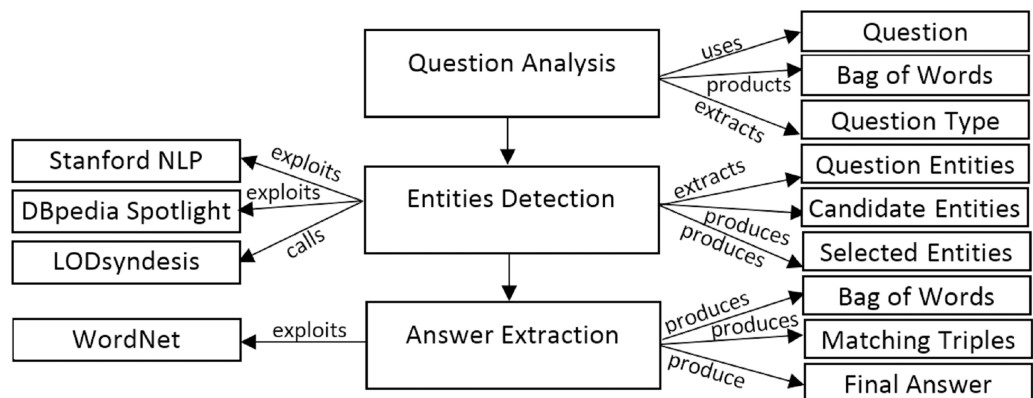

**Figure 4 The architecture of LODQA (*Dimitrakis et al., 2020*).**

    Since in this study the aim is to discover a suitable data source to extract the answer to the given question, so we will examine its application in the question answering system. Recently, the LODQA question answering system (*Dimitrakis et al., 2020*) uses the LODsyndesis (*Mountantonakis, 2018*) content-based data integration method to identify appropriate datasets. Figure 4 shows the architecture of the LODQA system.

    The proposed method for extracting the question schema is based on BERT and through LSTM. To compare hybridSP in the QA process, a KBQA based on BERT is used (*Su & Yu, 2020*). This system is called bertQA for short in this research. Figure 5 shows the bertQA system architecture. The two boxes on the right and left (with dashed line) run in the offline section and prepare the models needed for the online section.

## APPROACH

The proposed approach, hybridSP, addresses the challenge of source prediction in the context of several KB datasets and a corpus. The purpose of source prediction in current research is to find the best source containing the answer. Therefore, the hybridSP in the question answering process is presented in Fig. 6. The box (with dashed line) on the left
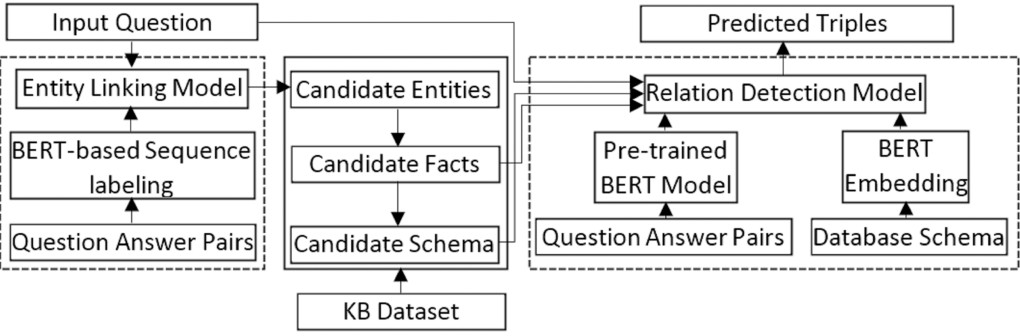

**Figure 5 BERT-based KB question answering system (bertQA) (*Su & Yu, 2020*).**

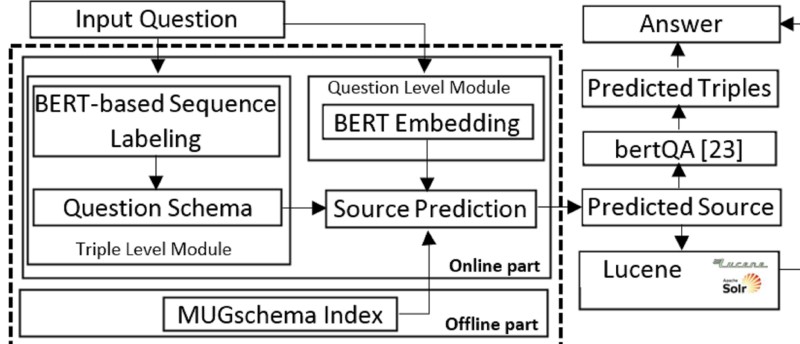

**Figure 6 The proposed system, hybridSP, architecture for source prediction (left blue box) in the response extraction process.**

of Fig. 6, which includes offline and online parts, shows hybridSP. If sources from the KG are predicted, the proposed method uses the bertQA system (*Su & Yu, 2020*) to search for answers. If the text source is identified as the source containing the answer, the Lucene is used to extract the answer.

Details of the hybridSP are described below.

## Offline part

The offline section, MUGschema index, provides the required platform for the online part. The main task in the offline part is to store the schema of the KB datasets so that in the online part, the process of predicting the response source is done at high speed. Figure 7 shows the tasks required in the offline part, which include the following steps.

### *Generating the mediated upper global schema*

Figure 8 illustrated the structure of creating an intermediate layer in the proposed method, taking ideas from data integration methods (*Ekaputra et al., 2017*). The production of the mediate layer includes the following steps: (1) categorizing the datasets according to the domain under study, (2) collecting the local schema of the datasets in each domain, (3) integrating the local schemas of each domain, and (4) integrating the global domain schemas and creating the mediated upper global level schema (the upper level

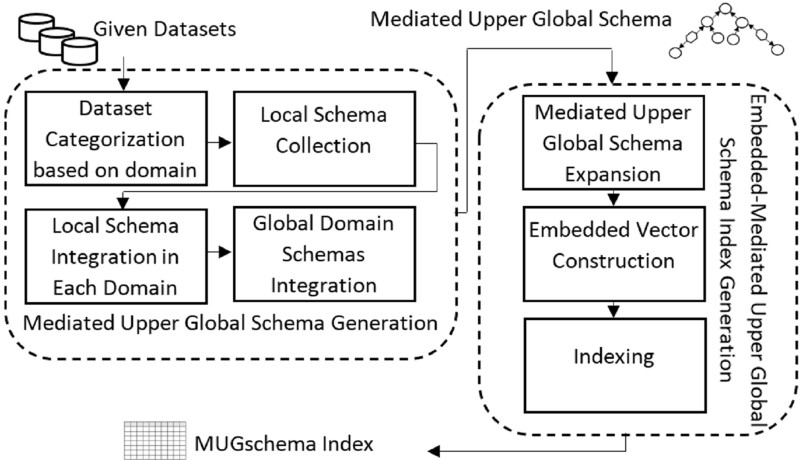

**Figure 7 Offline part steps.**

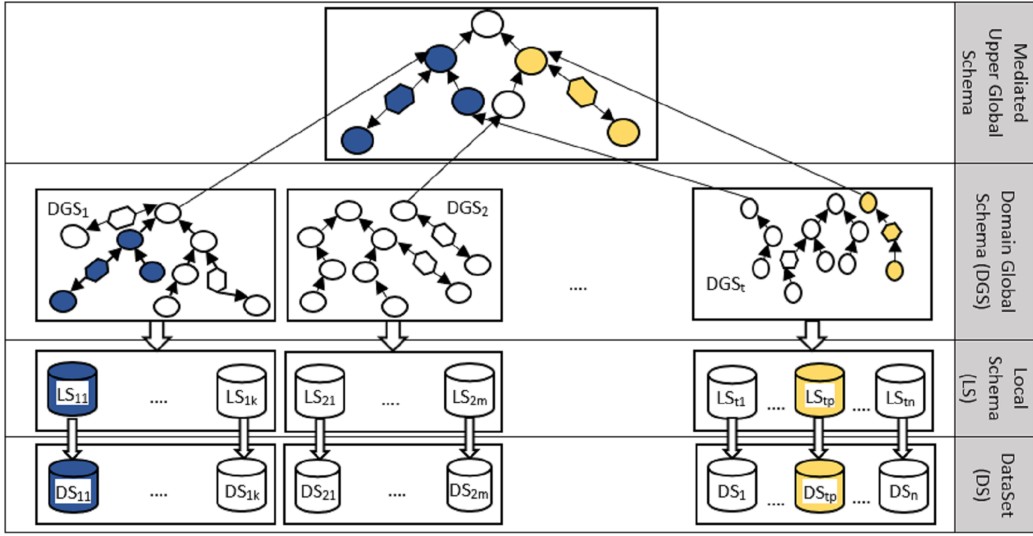

**Figure 8 The mediated upper global schema, MUGschema, generation process.**

in Fig. 8). Part of the mediated upper global schema generated between the three biomedical datasets (http://qald.aksw.org/index.php?x=task2&q=4), including Drugbank, Sider, and Diseasome, is shown in Fig. 9.

### Generating an embedded-mediated upper global schema index

This step consists of the following three minor steps. The first step is Mediated upper global schema expansion, which in this section for each of existed triple in the form of (object, predicate, subject) (*e.g.*, T1 in Fig. 10) that have an object neighbor triple (*e.g.*, $T_2$ in Fig. 10) and a subject neighbor triple (*e.g.*, $T_3$ in Fig. 10), three types of sequences $T_2 + T_1$, $T_1 + T_3$ and $T_2 + T_1 + T_3$ are produced. The purpose of expanding the generated schema in last step, is to increase the efficiency of the system for multi-hop questions, because to find the correct answer to such questions in KBs, it is necessary to infer several

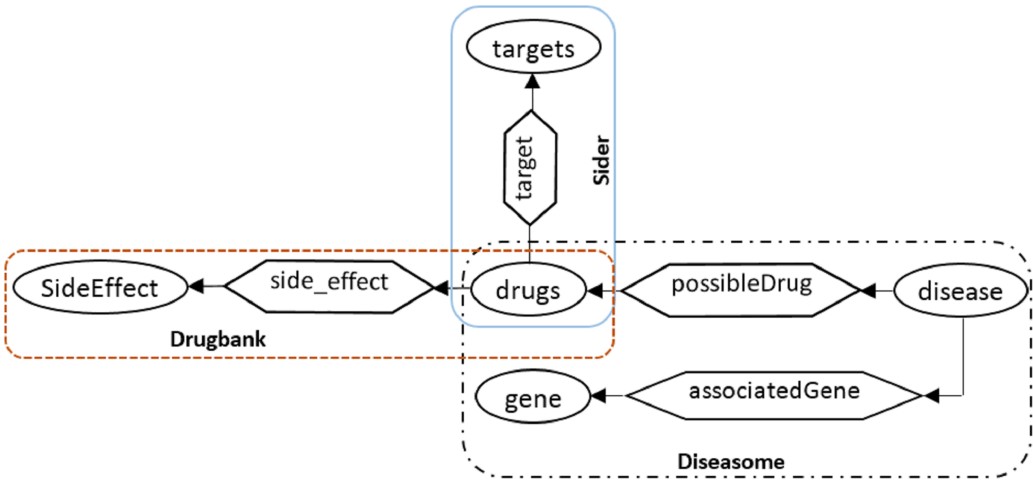

**Figure 9** An example to illustrate the mediated upper global level schema considering three biomedical datasets Drugbank, Diseasome and Sider.

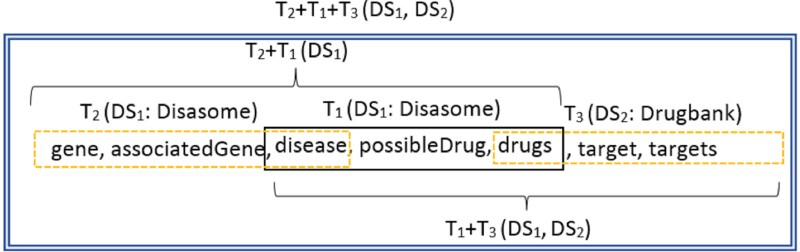

**Figure 10** An example to illustrate the triple expansion using neighbors.

neighboring triples. The Table 2 contains a 1-hop question and two types of multi-hop questions. The first question, $Q_1$, is a 1-hop and can only be answered with a triple. The second row shows a question, $Q_2$, belonging to the MetaQA (*Zhang et al., 2018*) database, which can be answered by inferring several neighboring triples from one KB dataset. The third row, $Q_3$, is a question from the biomedical QALD (*Wasim, Waqar & Usman, 2017*) dataset that to answer the question, triples from two different KB datasets must be considered. Both questions are 3-hop and require inference on three triples to arrive at the correct answer. The expansion of each triple with its neighbors will increase the accuracy of the system in selecting the appropriate sources containing the answers to the multi-hop questions.

Since the generated schema is the result of the integration of several schema datasets, the neighboring triples that merge with each other for expansion, may belong to different datasets that must be considered in the indexing step. According to the Fig. 10, it can be seen that $T_1$ and $T_3$ triples belong to Disasome and Drugbank datasets, respectively, so the resulting sequence $T_1 + T_3$ belongs to both datasets.

After expanding triples, the embedded vector of the generated sequences is created in second step and stored in the index file, MUGschema index, along with the identifier of
**Table 2** Examples of simple question (1-hop) from SimpleQuestions (*Bordes, Chopra & Weston, 2015*) and 3-hop questions from the MetaQA (*Zhang et al., 2018*) and biomedical QALD (*Wasim, Waqar & Usman, 2017*) datasest along with the triples needed to inference the relevant sources containing the answer.

$Q_1$: Which forest is Fires Creek in?
**T1**: (fires creek, containedby, nantahala national forest) – Freebase KB

$Q_2$: what types are the films directed by the director of "For Love or Money"?
$T_1$: (genre, type, film) – Freebase KB.
$T_2$: (Film, directedBy, person) – Freebase KB.
$T_3$: (person, director, movie) – Freebase KB.
$T_1 + T_2 + T_3$: (genre, type, film, Film, directedBy, person, person, director, movie) – Freebase KB.

$Q_3$: which genes are associated with diseases whose possible drugs target Cubilin?
$T_1$: (gene, associatedGene, disease) – Diseasome KB.
$T_2$: (disease, possible drugs, drugs) – Diseasome KB.
$T_3$: (drugs, target, targets) – Sider KB.
$T_1 + T_2 + T_3$: (gene, associatedGene, disease, possible drugs, drugs, target, targets) – Diseasome and Sider KBs.

**Table 3** A piece of generated index for MUGschema index based on extended triples in Fig. 10.

| Dataset ID | Embedded triple schema |
|---|---|
| 1 | $\overrightarrow{T_1}$ |
| 1 | $\overrightarrow{T_2}$ |
| 2 | $\overrightarrow{T_3}$ |
| 1 | $\overrightarrow{T_2} + \overrightarrow{T_1}$ |
| 2 | $\overrightarrow{T_1} + \overrightarrow{T_3}$ |
| 1, 2 | $\overrightarrow{T_2} + \overrightarrow{T_1} + \overrightarrow{T_3}$ |

the related datasets in third step. Table 3 shows part of the index stored for the information provided in Fig. 10.

## Online part

The present study employs two main modules: one, in a triple level, to extract features for comparison with LOD datasets (with an schema in mind) and one, in a question and word level (with content in mind) to extract features for comparison with the corpus.

These two modules are combined with other modules, Similarity Calculation, Binary Classification, and Multi-Labeled Classification, to predict the source(s) of the answer defined in the following discussion.

To select the source(s) of the answer, the current work proposes two variations containing two main modules, triple level and question level, in a serial or parallel combination with each other and other modules. In the first variation, the triple level module works in parallel with the question level module. However, in the second variation, the triple level module works in a serial combination in addition to a parallel combination and its output is concatenated in the question level module. Figures 11 and 12 present the graphical representation of the two proposed variations.

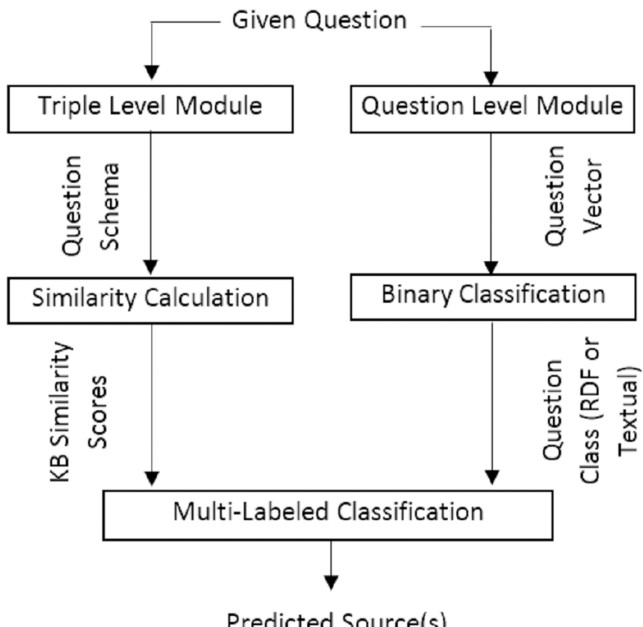

**Figure 11 The first variation of hybridSP in online section to predict the answer source(s).**

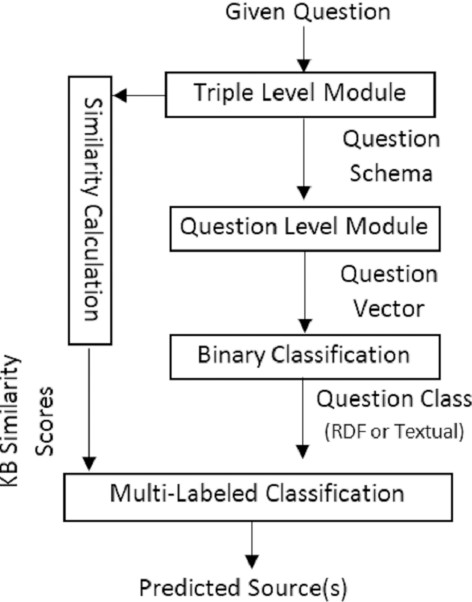

**Figure 12 The second variation of hybridSP in online section to predict the answer source(s).**

### Triple level module

This section introduces a question schema generation model for dataset prediction. The question schema, *e.g.*, (person, parent, person), carries very useful information for comparison with the dataset schema. The inspiration of this model is the work by *Yavuz et al. (2016)*, which proposes using the LSTM network method to extract the answer type.

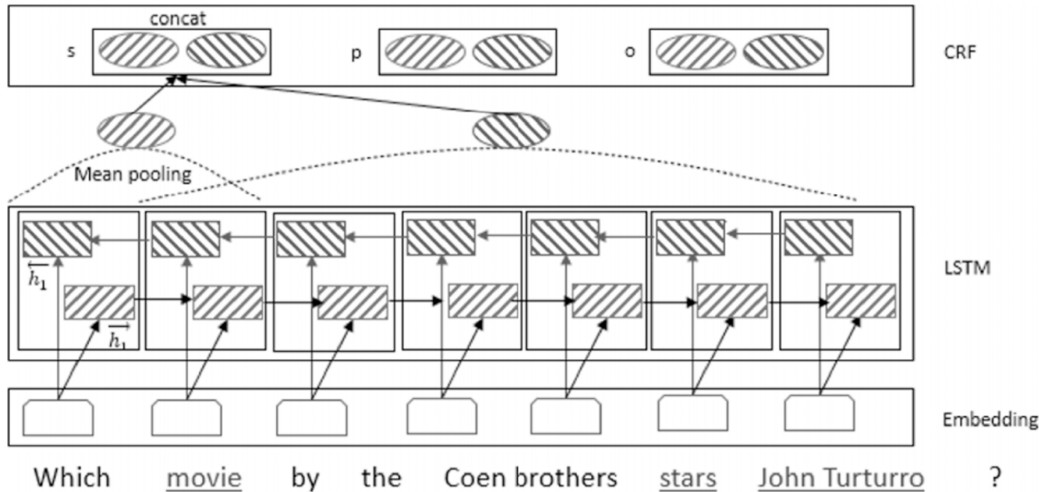

**Figure 13 The proposed question schema generation model in online section.**

**Decomposing question and generating question triple:** The first step in the Triple Level module is question decomposition and converting each sub-question into triple form (subject, relation, object) for use in the next step, Generating Question Schema.

For question decomposition and triple generation, the existing syntax and pattern-based work (*Shin & Lee, 2020*) are employed. After generating the triple of the given question, the current work's objective is to construct the schema of the triple. Sequence tagging is used to solve the question schema construction problem.

The key idea is to view each predicate/entity as a low-dimensional vector so that the relation information in KB may be retained. The proposed solution, therefore, integrates the low-dimensional representation in the first layer in order to transform a term from a character series into a dense vector sequence. Then, bi-LSTM (*Yavuz et al., 2016*) and CRF are used for producing the query schema to predict the datasets containing the answer. With such combining layers, past and future tags can be efficiently used to predict the class of the object and subject of the question.

**Generating question schema:** In order to reach a question schema, a confidence score for each possible KB type of the generated object, predicate, and subject must be computed *via* the bidirectional LSTM and CRF approaches (*Yavuz et al., 2016*). Figure 13 illustrates the proposed model for question schema construction.

It is supposed that the set of all class types in the underlying KB(s) is $T = (y_1, y_2, \ldots, y_k)$. The given mentions are the question words, $q = \{x_1, x_2, \ldots, x_n\}$, and the generated subject, predicate, and object from the last step are $s$, $p$, and $o$, respectively. For the given mention, $m$, with the starting $i$ and ending $j$ indices, the proposed approach, inspired by (*Yavuz et al., 2016*), employs an average pooling layer to generate the type of the mention. In the proposed model of the schema construction in Fig. 13, the hidden layer in the LSTM step computes the score matrix, $S_{3 \times k}$, in which three rows are for the mentions of subject, predicate and object. $s_{i,k} \in R$ denotes the likelihood of $y_k$ as the correct type of $m$ in $q$.

At first, LSTM is learned from left to right and right to left over embedding vectors of tokens, while the forward and backward outputs, $\overleftarrow{h}$ and $\overrightarrow{h}$, are computed. For all output nodes, $r \in [i, j]$, the backward $\overleftarrow{v_r}$, forward $\overrightarrow{v_r}$, backward $\overleftarrow{v_m}$, and forward $\overrightarrow{v_m}$ for mention $m$ and the final representation of $v_m$ (denoted for concatenation by $\oplus$) are obtained by the following equations:

$$\overleftarrow{v_r} = \frac{1}{r} \sum_{i=1}^{r} h_i \tag{1}$$

$$\overrightarrow{v_r} = \frac{1}{n - r + 1} \sum_{i=r}^{n} h_i \tag{2}$$

$$\overleftarrow{v_m} = \frac{1}{j - i + 1} \sum_{r=i}^{j} \overleftarrow{v_r} \tag{3}$$

$$\overrightarrow{v_m} = \frac{1}{j - i + 1} \sum_{r=i}^{j} \overrightarrow{v_r} \tag{4}$$

$$v_m = \overleftarrow{v_m} \oplus \overrightarrow{v_m} \tag{5}$$

The obtained score matrix, $S_{3 \times k}$, including the output nodes for the three mentions, $v_s$, $v_p$ and $v_o$, in sentence x enters the CRF layer to receive the probability distribution of class types. CRF is based on the graph, $G = (V, E)$, in which, V is a set of random variables, $Y_i$, E is an edge between $Y_{i-1}$ and $Y_i$, $f_k$ is a binary feature function, k is the number of defined edge variables, and $k'$ is the number of vertices and the two factors considered for vertices $V$, $F_V$ and edges $E$, $F_E$ (Eqs. (6) and (7)). S(x, y) is the sum of the two factors (Eq. (8)) (x is the sequence of tokens $v_s$, $v_p$, and $v_o$, and y is the sequence of tags). The conditional probability p(y|x), is obtained by Eq. (9). The goal is to maximize the probability of the distribution function results in $y^* = \text{argmax}_{y' \in y}(s(x, y'))$.

$$F_V = \sum_{i=1}^{3} \lambda'_k f'_k(y_i, x) \tag{6}$$

$$F_E = \sum_{i=1}^{3} \lambda' k f'_k(y_{i-1}, y_i, x) \tag{7}$$

$$s(x, y) = F_V + F_E \tag{8}$$

$$p(y|x) = \frac{e^{s(x,y)}}{\sum_{y' \in y} e^{s(x,y')}} \tag{9}$$

### Question level module
For all words, $w_i$, in the given question, $q_i$, a vector of features, $V_{wi}$, is generated. Two approaches are proposed:

1. Language feature vector (LF): This vector features properties about parts of speech tagging (POS), dependencies, and the semantic role labeling of each word, $w_i$.

2. Embedding feature vector (EF): These word embedding vectors are pre-trained using BERT on Wikipedia pages and BookCorpus. For each word, a 768-dimensional vector is obtained. The question vector is computed with the average function over the word vectors.

### Similarity calculation module

For each underlying KB, G, the schema set, $S = \{s_1, s_2, \ldots, s_n\}$, is prepared based on the definition in "Question Schema". The average embedding vectors for the schema set $S_e = \{s_{e1}, s_{e2}, \ldots, s_{en}\}$, and the given question schema, $v_{qs}$, are calculated with the average function, AVG in Eq. (10). A maximum reverse distance, D to $sim(v_{qs}, G)$, is assigned in the Eq. (11).

$$v_{qs} = AVG(v_s, v_p, v_o) \tag{10}$$

$$sim(v_{qs}, G) = max_{se \in S_e} \frac{1}{D(v_{qs}, se)} \tag{11}$$

### Binary classification module

The binary classification module classifies the vector of the given question based on its textual features in the RDF or textual classes.

### Multi-label classification module

In previous modules based on the applied model, two categories of data are extracted: (1) calculated similarity scores, $SS = \{sim_{G1}, sim_{G2}, \ldots, sim_{Gm}\}$, for m underlying KBs, which determine the similarity score of the question schema to underlying KBs and (2) the class of the source of the answer (structured (RDF) or unstructured (textual)). In this module, the aim is to combine textual and underlying KB features to predict the sources(s) containing the answer. The current research assumes that its underlying sources are one or more knowledge graph (s) and one corpus. In the context of a hybrid search, the answer of the given question is extracted using one or more source(s), considered as the question class labels. As more than one class label for each given question is considered, a multi-labeled classification (MLC) is a good fit as the solution for the source prediction problem allowing multiple predictions for several classes at the same time.

## EXPERIMENTS

This section describes the benchmarks, data preparation, and experimental results. Performance is evaluated in terms of the three main tasks contributed by the proposed approach: (i) question schema extraction (triple level), (ii) binary or question level classification, and (iii) source prediction (a combination of the triple level and question level). In his study, the purpose of source selection is to find suitable sources for extracting answers. Therefore, the impact of the presence of source prediction method in the QA system is finally evaluated. All experiments are conducted on a machine with Nvidia Tesla v100 and p100 GPUs.

The accuracy is reported as the evaluation metric. Accuracy indicates the ratio of the number of correctly identified samples to the total number of samples. For each question, $q_i$, let $T$ be the true set of labels and $S$ be the predicted set of labels. Accuracy is measured by the Hamming score which symmetrically measures how close $T$ is to $S$ (Eq. (12)).

$$Accuracy(q_i) = |S \cap T|/|S \cup T| \qquad (12)$$

## Benchmarks

To assess the tasks in the literature's related works, two types of benchmarks are employed: (i) benchmarks used for question answering systems based on KBs, such as simpleQuestions (*Bordes, Chopra & Weston, 2015*), webQuestions (*Xu et al., 2016*), webQuestionsSP (*Sun, Bedrax-Weiss & Cohen, 2019*), ComplexQuestions (*Bao et al., 2016*), ComplexWebQuestions (*Talmor & Berant, 2018*), WikiMovies (*Miller et al., 2016*), MetaQA (*Zhang et al., 2018*), LC-QUAD, and LC-QUAD 2.0 (*Fu et al., 2020*), multi-hop Biomedical datasets in QALD challenge (http://qald.aksw.org) (*Wasim, Waqar & Usman, 2017*) and (ii) benchmarks used for hybrid question answering systems, such as SPADES (*Das et al., 2017*) and hybrid dataset in QALD challenge (*Usbeck et al., 2018*).

For its evaluation of schema extraction, question level classification, and source prediction, the current work employs the ComplexWebQuestions, which is constructed over webQuestions with additional constraints and SPARQL logical forms. WebQuestionsSP and ComplexQuestions are also constructed over webQuestions and WikiMovies has the same properties as webQuestions. Thus, the present study does not consider these benchmarks. LC-QUAD is a benchmark in the current study and used in QAnswer (One of the basic QA systems in present research) as well. The three tasks are evaluated against hybrid QALD benchmark. In the section of source prediction evaluation as well as answer extraction, three categories of simple questions containing SimpleQuestions (*Bordes, Chopra & Weston, 2015*), multi-hop questions containing biomedical QALD (*Wasim, Waqar & Usman, 2017*) and MetaQA (*Zhang et al., 2018*) and hybrid questions containing hybrid QALD (*Usbeck et al., 2018*) and SPADE (*Das et al., 2017*) benchmarks are used.

Table 4 lists the general characteristics of the benchmarks used.

## Data preparation

Since the current study's approach is based on schema dataset integration, the schema of the underlying knowledge graphs (s) must be determined. As seen in Table 4, the benchmarks are related to DBpedia, Freebase, Wikidata, and CluWeb, Sider, Diseasome, Drugbank and WikiMovies. Additionally, for the response time evaluation and comparison with QAnswer, the questions are executed over DBpedia, Wikidata, Freebase, MusicBrainz, and DBLP (*Diefenbach et al., 2019*; *Diefenbach et al., 2020*). As a result, the RDF data of underlying KBs are collected in the form of triples (s, p, o) and then the type of subject s and object o are retrieved to prepare the schema of the KGs.

**Table 4 General characteristics of benchmarks used in the evaluations.**

| Benchmark | Size | Type of background | Source background |
|---|---|---|---|
| SimpleQuestions (*Bordes, Chopra & Weston, 2015*) | 1,000 | KB | FreeBase |
| LC-QULD (*Fu et al., 2020*) | 5,000 | KB | DBpedia |
| hybrid QALD (*Usbeck et al., 2018*) | 550 | Hybrid | DBpedia, Wikipedia |
| biomedical QALD (*Wasim, Waqar & Usman, 2017*) | 50 | KB | Sider, Diseasome, Drugbank |
| ComplexWebQuestions (*Talmor & Berant, 2018*) | 34,689 | KB | FreeBase |
| MetaQA (*Zhang et al., 2018*) | 39,282 | KB | WikiMovies (*Miller et al., 2016*) |
| SPADES (*Das et al., 2017*) | 43,000 | Hybrid | FreeBase-CluWeb |

The state-of-the-art approach, hMmatcher (*Yousfi, El Yazidi & Zellou, 2020*), performs the schema matching against schema.

## Training data

Since the considered databases only provide question-answer pairs, it is necessary to figure out the type information to be used as training data. Simulated types are obtained as follows: for each relation $r$, $(?, r, o)$ and $(s, r, ?)$ are queried based on the underlying KGs. The subject and object types are selected as related types. In this research, the PyTorch (https://github.com/huggingface/pytorch-transformers) implementation of BERT from Huggingface has been used to encode the questions and POS tags. In the pre-training, the bert-base-uncased model (*Devlin et al., 2019*) is used. This model includes 12 transfer blocks, hidden layer with size 768, 12 attention layers and Adam optimizer with 110 M parameter. After the vectors are obtained by BERT, for fine-tunning step a linear layer for binary classification and several layers for multi-label classification stacked on top of the BERT output are applied. In multi-label classification, the output of the model will be the probability of all classes. For fine tuning, a learning rate of 3e−5, a batch size of 128 and 4 epochs is used.

## Question schema extraction

For the first task of question schema extraction, the current study utilizes 768-dimensional word and parts of speech (POS) embeddings, which are pre-trained using BERT on Wikipedia pages and BookCorpus. The size of the LSTM hidden layer is set at 768 in a forward and backward direction. A 768-dimensional vector representation is learned for each class of subject and object. RMSProp (*Zou et al., 2019*) is utilized with a learning rate of 0.005. Because the dropout method is an efficient way of mitigating over fitting, the present research applies it to the embedding, LSTM, and CRF layer outputs. The dropout rate is 0.2 and the hyper-parameters are selected on training data based on cross-validation. The networks are implemented on the common Theano neural network toolkit (*Erickson et al., 2017*).

For each given question, the evaluation relies on two annotators to judge the generated schema. If the output of the two annotators is correctly labeled, then this is considered as the correct output. The agreement rate (Cohen's kappa) is 0.80%.

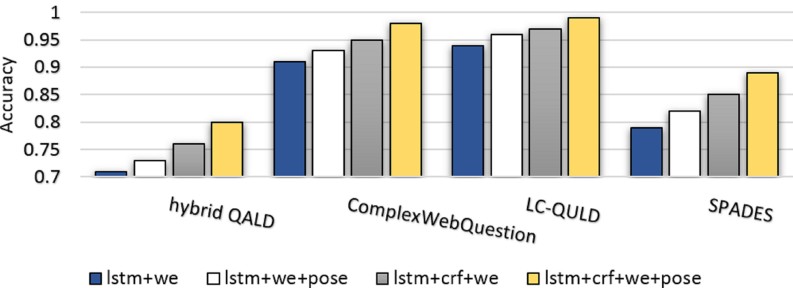

**Figure 14 Comparison of accuracy metric among several benchmarks against different applied settings in question schema extraction.** Denotations *lstm*, *crf*, *we*, and *pose* indicate use of the LTSM approach, CRF approach, word embeddings and P.     

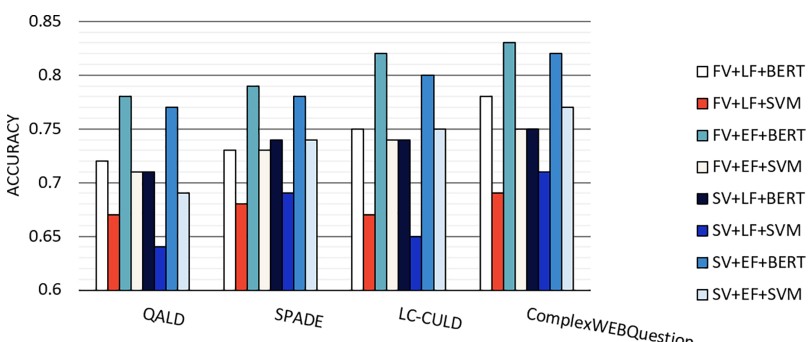

**Figure 15 The accuracy for classifying questions in the KG or text classes.** Results are calculated by applying the first variation (*FV*), second variation (*SV*), embedding feature vectors (*EF*), language feature vectors (*LF*), BERT and SVM class.     

Figure 14 illustrates the comparison of the accuracy metric among several benchmarks against different applied approaches. As seen, ComplexWebQuestion and LC-QULD have a higher accuracy because their questions are selected from KG (s). The schema of such questions which corresponds to the schema of underlying KGs are extractable. Hybrid benchmarks (hybrid QALD and SPADES) include questions that can only be answered from the text. The schema of these questions is not available in the KG. Furthermore, the use of CRF, word embedding, and POS embedding leads to improved performance.

## Question level classification

The second case is binary or question level classification. The only system that refers to the choice of source between a KG and a text source is the HAWK system (*Usbeck et al., 2015*). In this system, noun phrases are searched from the text and the rest from the KG which no evaluation has been provided for it. In this section, first, we will evaluate the binary class with two tags, text and RDF, on different datasets and based on different settings. Then we compare the hybridSP with the HAWK method.

The constructed BERT vector of the given question is classified based on a model fine-tunning. This model employs a binary classifier layer, to classify the question in either two classes of, *i.e.*, KG or corpus class. Figure 15 provides the accuracy for classifying

**Table 5 Categories found by clustering question embedding vectors from the text class.**

| Cluster | Examples containing words |
|---|---|
| Sequence Adverbs | first, second, after, former |
| Adverbials of Location | around, front, among |
| Event Indicators of Time | during, when, the time of |
| Quality Adjectives | most, largest, deeper |
| Adverbs of Time | early, still, yet, already |
| Quantifiers | only, some, all |
| Adverbs of Frequency | often, sometimes, usually |
| Hedges | probably, definitely, certainly |

questions. Results are calculated by applying the first variation (FV) (Fig. 11) or the second variation (SV) (Fig. 12) of the proposed model and embedding feature vectors (EF) or language feature vectors (LF) and the BERT classifier or SVM classifier. Scikit-learn 0.19.2 is used for SVM and PyTorch for BERT classifier. As seen in Fig. 15, the large benchmarks show a better performance than those by the smaller benchmarks. In addition, between the two large benchmarks and the two small ones, the benchmarks constructing questions over KGs show a better performance than the hybrid benchmark.

As shown in previous research (*Gonz & Eduardo, 2005*), SVM is less efficient than BERT. In terms of comparing the first and second model variations, one can see that the second model's performance is better. While the binary classifier in the first model only works based on the question level, the second model concatenates the triple level and question level as inputs to the classifier. Furthermore, the performance of embedding feature vectors surpasses that of language feature vectors. A possible explanation for this is the consideration of both semantic and syntactic features when using embedding space.

The question classification in the hybridSP and the HAWK system on the QALD benchmark had an accuracy of 0.78 and 0.45, respectively. These values on the large SPADE benchmark showed 0.79 and 0.56, respectively.

To extract the properties involved in categorizing questions in the text category, the present study runs a clustering algorithm over embedding vectors of the questions. With an analysis of each cluster by a natural language expert, the categories in the first column of Table 5 are recognized. These categories include *Sequence Adverbs, Adverbials of Location, Event indicators of Time, Quality Adjectives, Adverbs of Time, Quantifiers, Adverbs of Frequency*, and *Hedges*. In the second column, a number of examples are given for each category. By comparing the embedding vectors of other words with the embedding vectors of given examples, the other items in these categories are obtained.

The study of the extracted categories and the identification of more categories through the examination of other question benchmarks should be left to future research. One can refer to *Al-Khawaldeh (2019)*, *Ulinski, Benjamin & Hirschberg (2018)* for further work in this area.

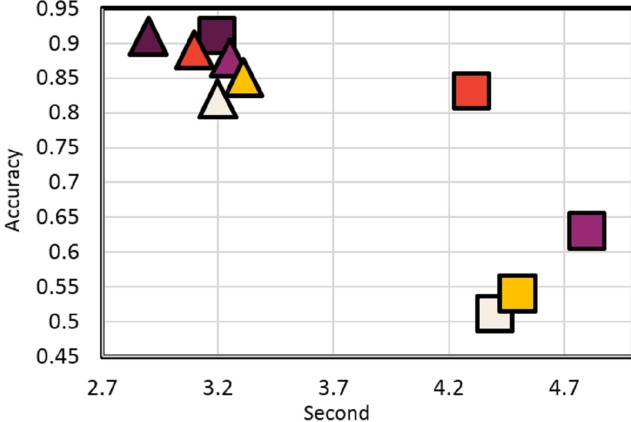

**Figure 16 Comparison of hybridSP and LODsyndesis based on the accuracy-prediction time trade-off.**

## Source prediction

In this section, we first compare hybridPS with the HAWK method, for predicating the source. Then we will compare hybridPS with the most similar method, LODsyndesis, that selects the source on the KG sources.

The source prediction in the hybridSP and the HAWK system on the QALD benchmark had an accuracy of 0.82 and 0.45, respectively. These values on the large SPADE benchmark showed 0.85 and 0.56, respectively. Examining the result of this evaluation with the previous section (Question Level Classification) shows the effect of considering the schema of the question on the accuracy.

As mentioned in the previous sections, in predicting the appropriate source in KGs, the most similar method to the hybridSP is the LODsyndesis system. This system selects the sources available in the KG by focusing on the entity provided. HybridSP, in addition to KG sources, also predicts the existence of responses in the textual source. It also provides solutions for multi-hop questions to increase source predicting efficiency. Therefore, in addition to evaluating the two systems on simple questions, multi-hop and hybrid benchmarks has also been used to compare the two systems. By referring to Fig. 16, it can be seen that both systems have the same accuracy in relation to simple questions.

Unlike LODsyndesis, which is based on triples, hybridSP searches based on the schema of triples. The schema of triplets has a much smaller number of triples. Thus, for all benchmarks, source prediction in LODsyndesis takes longer than in hybridSP.

Two different types have been used to evaluate the two systems on multi-hop questions. In the first type (*e.g.*, MetaQA), the triples related to each question are present in a common KG dataset, while in the second type (*e.g.*, biomedical QALD), they are present in two or more different datasets.

As can be seen, the LODsyndesis system offers much less accuracy than the hybridSP system because it does not infer multiple sequential relationships. The reduction in accuracy is greater in relation to the second type. This observation is due to the fact that in the first type, the question entity and the answer are located in a common KG, while in the

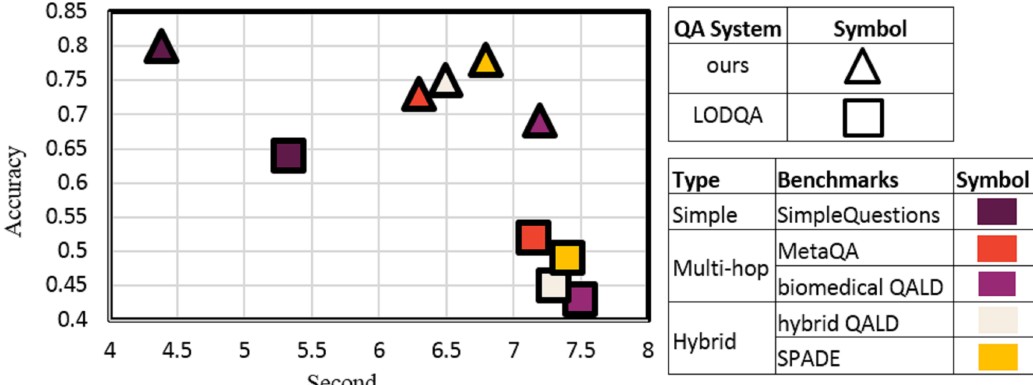

**Figure 17 Comparison of BERT-based QA (*Su & Yu, 2020*) and hybridSP method integration with LODQA (*Dimitrakis et al., 2020*) using LODsyndesis (*Mountantonakis, 2018*) based on the accuracy-response time trade-off.**

second type, the two are often in different KGs. LODsyndesis focuses on entity-based triples in each KG, thus it will be less efficient in case of such questions.

## The efficiency of the hybridSP in the QA system

In this research, the Question answering benchmarks has been used to evaluate the hybridSP to select the appropriate source containing the answer. Therefore, in this study, we also evaluate the efficiency of the hybridSP in the QA system. In this section, we first compare the accuracy and average response time of two QA systems, the former using hybridSP and the latter using the LODsyndesis method (LODQA (*Dimitrakis et al., 2020*)) to select the appropriate source. Then we will examine the result of adding hybridSP to the QAnswer system in the difference of response time.

As shown in Fig. 16, the hybridSP system is more efficient than the LODsyndesis system in terms of accuracy and prediction time. In this section, we will evaluate the role of two systems in the answer extraction process. Recently, the LODQA system has used LODsyndesis to identify the source. Because LODQA uses traditional methods in question representation to extract answers, this study uses powerful BERT-based question answering system (*Su & Yu, 2020*) in its evaluation.

As can be seen in Fig. 17, the accuracy of the proposed method has been significantly improved compared to LODQA due to the presence of hybridSP alongside the powerful BERT-based QA system. The remarkable thing about Fig. 17 is the low accuracy of the BERT-based QA system for multi-hop questions. The reason is that this system works in a single relational category and does not deal with multi-hop questions.

We compare the response time within the QAnswer framework with and without the source prediction. In QAnswer, questions are executed over DBpedia, wikidata, Freebase, MusicBrainz, and DBLP. The QAnswer system reports an average response time of 1.37 s for the hybrid QALD benchmark when executed over DBpedia. If this is executed on all underlying datasets (DBpedia, wikidata, Freebase, MusicBrainz, and DBLP), the need to run QAnswer on all datasets was eliminated and only on predicted resources containing responses. With the addition of hybridSP alongside QAnswer, the average

response extraction time was reduced to 5.8 s. The average of source prediction time by hybridSP on the stated datasets is 4.11 s. While the response time average by QAnswer on the predicted sources is 1.69 s. The increase of about 0.3 s compared to the QAnswr runtime on DBpedia is due to the fact that for some questions, sources in addition to DBpedia containing the answer has been identified.

## CONCLUSION AND FUTURE WORK

To extract a final answer, a question answering committee has recently been presented to hybrid approaches by integrating structured (KG) and unstructured (corpus) sources. Also, many studies have set out to extract the response from LOD. However, the majority of the work already conducted on datasets which are pre-determined and limited. In QA on a large number of sources, it is very important to choose the relevant sources to maintain scalability. So far, no method has been proposed for source selection in hybrid sources. In KG, the data integration domain chooses the source among KGs. In this study, a data integration-based method with a content-based mediation approach is presented. The proposed method, hybridSP, includes three main tasks of (i) generating a question schema (triple level), (ii) question classification (word and question levels) and (iii) combining the results of the first and second parts in identifying the source containing the answer. The hybridSP aims to identify the source containing the answer. Therefore, in this research, the hybridSP for selecting the appropriate source along with the QA system as a general task has also been investigated. The mentioned tasks were evaluated using several benchmarks from three categories of simple, multi-hop and hybrid questions. The evaluation results showed that the proposed system of source selection compared to the methods of source selection from hybrid sources, HAWK, as well as KG sources, LODsyndesis, have a higher efficiency in accuracy and execution time. For the general QA task, the BERT-based QA system was used along with the proposed source selection system. A comparison of the combination of the two systems found a significant increase in performance compared to the LODQA system, which uses LODsyndesis. Moreover, the addition of hybridSP alongside QAnswer system significantly reduces the average response time. QAnswer has the ability to respond in the context of several KG datasets. Future research shall extend the scope of this work by considering a larger number of KGs and textual resources over large-scale complex questions especially multi-hop benchmarks. Also, index clustering containing BERT embedded vectors to reduce the prediction time is considered in future work.

### Funding

The authors received no funding for this work.

### Competing Interests

The authors declare that they have no competing interests.

## Author Contributions

- Somayeh Asadifar conceived and designed the experiments, analyzed the data, performed the computation work, prepared figures and/or tables, and approved the final draft.
- Mohsen Kahani performed the experiments, authored or reviewed drafts of the paper, and approved the final draft.
- Saeedeh Shekarpour conceived and designed the experiments, authored or reviewed drafts of the paper, and approved the final draft.

## Data Availability

The code is available in the Supplemental Files.

Third-Party data are available at:

- Benchmark: LC-QULD, Size: 5000, Type of Background: KB, https://github.com/AskNowQA/LC-QuAD.

- Benchmark: QALD-6, Size: 550, Type of Background: Hybrid, http://qald.aksw.org/.

- Benchmark: ComplexWebQuestions, Size: 34689, Type of Background: KB, https://www.tau-nlp.org/compwebq.

- Benchmark: SPADES, Size: 43000, Type of Background: Hybrid, https://rajarshd.github.io/TextKBQA.

## Supplemental Information

Supplemental information for this article can be found online at http://dx.doi.org/10.7717/peerj-cs.846#supplemental-information.

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
