# Peer review of "Schema and content aware classification for predicting the sources containing an answer over corpus and knowledge graphs"

_PeerJ Computer Science, doi:10.7717/peerj-cs.846_

## Round 0.1 · original submission · Major Revisions

We received two consistent review reports. Please revise the paper and provide a detailed response letter. The experiment design and methodology should be strengthened.

·

Basic reporting

The article is well thought out, well-contextualized, and innovative. The references can be seen to provide sufficient background of the field, but there are some references that are relatively old. The research results in the QA field in recent years should be supplemented and compared. The text of the article is clearly structured and rich in content. However, as shown in Figure 2, the image descriptions are too simple, the diagrams are not clear, and the text is rather blurred. The terminology in the article should be more precisely explained and illustrated. In addition, the English language should be improved to ensure that an international audience can clearly understand your text.

Experimental design

The motivation behind the problem investigated in this manuscript is interesting and meaningful. However, the detailed application background of this issue has not been mentioned in detail. This article has a good grasp and investigation of the existing QA research, but lacks recent work comparison and narration. There are too many experiments using a combination of machine learning methods for comparison and too many repetitive experiments. It is recommended to compare with the existing state-of-the-art methods.

Validity of the findings

The data provided by the article is relatively reliable and abundant, and the conclusions are related to the original research questions. However, the article should emphasize the impact of the proposed method on the field and its novelty.

Additional comments

no comment

·

Basic reporting

- In section 1, the authors should introduce a model that did the work like this study before. How is the authors’ model different from the previous model?
- In section 3, the authors should define general metadata, detail metadata, and give some examples.
- Figure 2 should label on shapes and give some explanations.

Experimental design

- In lines 194-196, expansion of the existing triples aims to increase the efficiency of the system in answering multi-hop questions. The authors should give some examples of multi-hop answers and explain how the model finds the multi-hop answers from expansion triples.
- In lines 321-322, the authors used 150-dimensional word and parts of speech (POS) embeddings with the word2vec tool. As I know the word2vec model appeared in 2013. Why did the author not use fast text (2016) or BERT (2018)?
- In lines 329-330, The author hired two annotators to evaluate the question schema extraction task. Why don’t the authors use a computerized model instead of humans for objective judgment?
- In line 344, the authors used naïve Bayes and SVM binary model for question-level classification. Why don’t the authors use deep learning such as LSTM, CNN, or BERT for classification to achieve higher accuracy?
- In lines 372-374, the scores such as accuracy, precision, recall, and F1 should be presented in equation forms and numbered.
- In subsection 5.7, the authors should present tables that compare the response time of this study over datasets. These tables prove that the QAnswer framework achieves better results on two categories with and without source prediction.

Validity of the findings

No comment

Additional comments

This paper provides a solution that combine knowledge graph and text to predict the source (s) containing the response from a number of structured data sources and an unstructured data source. This is an interesting method.

---

## Round 0.2 · accepted · Accept

The paper can be accepted. Congratulations!

·

Basic reporting

The authors addressed all my previous comments.

Experimental design

The authors addressed all my previous comments.

Validity of the findings

The authors addressed all my previous comments.